# Leveraging PDDL to Make Inscrutable Agents Interpretable: A Case for Post Hoc Symbolic Explanations for Sequential-Decision Making Problems

**Sarath Sreedharan, Subbarao Kambhampati**

Arizona State University

### Abstract

There has been quite a bit of interest in developing explanatory techniques within the ICAPS-community for various planning flavors, as evidenced by the popularity of the XAIP workshop in the past few years. Though most existing works in XAIP focus on creating explanatory techniques for native planning-based systems that leverage human-specified models. While this has led to the development of valuable techniques and tools, our community tends to overlook a very important avenue where the XAIP techniques, particularly ones designed around symbolic human-readable models, could make a practical and immediate impact. Specifically, we can use them to generate symbolic post hoc explanations for sequential decisions generated through inscrutable decision-making systems, including Reinforcement-Learning and any inscrutable model-based planning/approximate dynamic programming methods. Through this paper, we will discuss how we could generate such post hoc explanations. We will also discuss how to use and adapt the current XAIP and model learning techniques to address many explanatory challenges within this realm and motivate some of the open research challenges that arise when we try to apply our methods within this new application context.

## 1 Introduction

Recent years have seen an upswing in the number of works done in the space of explainable planning (Chakraborti, Sreedharan, and Kambhampati 2020). Many strides have been made in introducing tools and techniques designed to explain decisions derived through automated planning methods. Though there remains a particular area of application, ripe for investigation from the XAIP community, that seems surprisingly underexplored: The use of symbolic models to generate post hoc explanations for sequential decisions derived from inscrutable systems. While there are some works like that by Sreedharan et al. (2020), which have started looking at pieces of the problem, the direction as a whole seems mostly overlooked. This is unfortunate given the vast array of techniques developed in our community that could be leveraged to address many explanatory challenges within these settings by first mapping the task information into a symbolic model description like PDDL. Moreover, models like PDDL provide a very intuitive representation for planning problems as they are built around concepts from folk psychology (Miller 2018). In the end, this work follows a growing consensus that while it is unclear whether AI systems themselves would need to use symbols in their internal processing for effective decision-making, there is no doubt that people are comfortable with and expect to communicate with these systems in terms of symbols that are meaningful to them. PDDL provides a particularly expressive, intuitive, and well-studied representation for sequential decision-making problems which could be used for such communication purposes.

In this paper, we hope to make a systematic case for applying and adapting existing methods for both explanation and model learning to this new setting. We will provide a rough characterization of potential solutions to this problem and possible scenarios where such solutions may fall short. We will end the paper with a discussion of open problems related to post hoc explanations, including the challenges related to ensuring the precision/soundness of explanations. A core tenet that underlies many of the most successful symbolic AI methods is the idea of compiling problems into an equivalent but more pliable representations. That is, we understand the same problem could be represented in multiple ways, and the choice of representation scheme in which to express the problem is one we are free to make to optimize for our ability to solve the problem and to reuse existing tools. It is time we bring the same ethos to explanation by recognizing that translation of black-box problems into symbolic models like PDDL provides us with many advantages, including the ability to draw from rich literature on explanation generation built around such models.

## 2 Case Study: Montezuma's Revenge

Following Sreedharan et al. (2020), we will use Montezuma's revenge (Wikipedia contributors 2019), a popular Reinforcement Learning (RL) benchmark that requires sequential reasoning, as an example to illustrate the potential utility of post hoc symbolic explanations. Figure 1 presents a screenshot from the first level of the game. Here the player is supposed to get the key on the left side of the screen. Let us imagine an observer who is not familiar with all the rules of the game is trying to make sense of plans generated by an automated decision-making system. Let us assume the system comes up with a plan that involves Panama-Joe (the protagonist of the game), making its way to the lowest platform, then moving left and then jumping over the skull, and finally

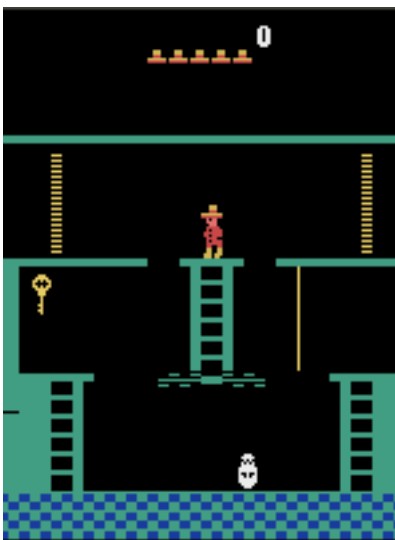

Figure 1: A screenshot from the first screen of the game Montezuma's revenge.

getting to the key. A user unaware of the effects of touching the skull may ask why the agent did not try to move through the skull. The system itself may not be reasoning over concepts like skulls, ladders, etc., but may instead be reasoning at the level of pixels or game RAM states. So it may be unable to communicate that touching the skull when Joe does not have a sword would end up killing him. The system could try to simulate this alternate plan (assuming the system can provide some visualization of the game) and if the system is reasoning over pixels, even try to highlight all the regions in the failure state that the system believes are important (which hopefully includes the skull and the inventory region). Unfortunately, in addition to increasing the cognitive load on the human's end, as pointed out by previous works like Atrey, Clary, and Jensen (2019) such visual annotations can be confusing. Instead, the user might prefer the agent to describe why this alternate plan may fail in terms they understand. For example, the agent could explain that for Joe to move left it needs to satisfy the precondition not_next_to_skull or has_sword. By doing this the system is effectively generating explanation not in terms of the model it used to come up with the original decision, but is instead using a different but an equivalent symbolic representation that is easier for the user to understand. Such use of post hoc symbolic models, allows the system to freely choose representation schemes best suited for its decision-making without worrying too much about potential explanatory overhead such choices may entail.

## 3 Post Hoc Symbolic Explanation

Figure 2, presents an overview of the learning/explanation generation process that could be used to generate such post symbolic explanations. The overall process starts with the user of the system providing a set of vocabulary items (i.e. labels for actions and state fluents that the user understands) and an explanatory query (for example a contrastive query

(Miller 2018)). Within the context of an AI system, the user described here could correspond to people interacting with the system under many capacities, including system designers, end-users, and even domain experts. In this paper, we will mostly be agnostic to the specific user types, though one could easily see the role and background of the user could change the type of approaches used in each step. Also in the most general cases, the one specifying the vocabulary and the one raising the query need not be the same, but we will also ignore this distinction for now. With the vocabulary set and the query in place, the explanation generation procedure can interact with the actual system to generate a symbolic approximation that is sufficient to provide a response to the current query. If the system fails to identify an appropriate explanation, this is a good indicator that the original vocabulary set is incomplete and would require additional vocabulary items to create a higher fidelity symbolic model to generate the required explanations. In the rest of the section, we will look at each of these individual steps of the overall flow. One of the requirements for this method is the ability to leverage the internal models of the agent. Here we are using the term model in a very general sense. These could correspond to learned models being used by the agent (for example neural network models learned over latent state representations), procedural models, internal simulators, or even non-parametric models consisting of original experience being used by the agent to form its policy. The only requirement we place on the model is that we are able to interact with it and potentially sample experiences from it. In regards to the symbolic model, we will generally assume some variant of PDDL.

### 3.1 Vocabulary Learning

One of the core research challenges we are trying to address here is that of vocabulary mismatch. Plainly put, we need to overcome the fact that the system may be reasoning about the task in terms that a user of the desired background may not understand. Thus, mapping these models into any symbolic representation isn't enough as these may still be defined in terms that carry no real significance to the users. After all, a single atomic transition system could be synthesized from quite a different set of PDDL models, defined over various state factors. This makes many of the automatic model synthesis methods like that from Bonet and Geffner (2019) or Asai and Fukunaga (2018), which also try to automatically generate symbols, ill-suited for our purposes. One of our core proposals in this paper is the need to include human input in some stage of the learning pipeline to determine the factors on which the model will be built. Basically, we would need people to specify at least parts of the action space and the state fluents over which the model would be learned. The core requirement for each vocabulary item is for the system to learn a way in which it can detect when the state contains a specific fluent value or when the action (or a trajectory) performed by the agent corresponds to an action specified by the user. One way to represent such mappings from system representations to human vocabulary items would be to learn binary classifiers for each item from data collected by interacting with the user. In general, all the

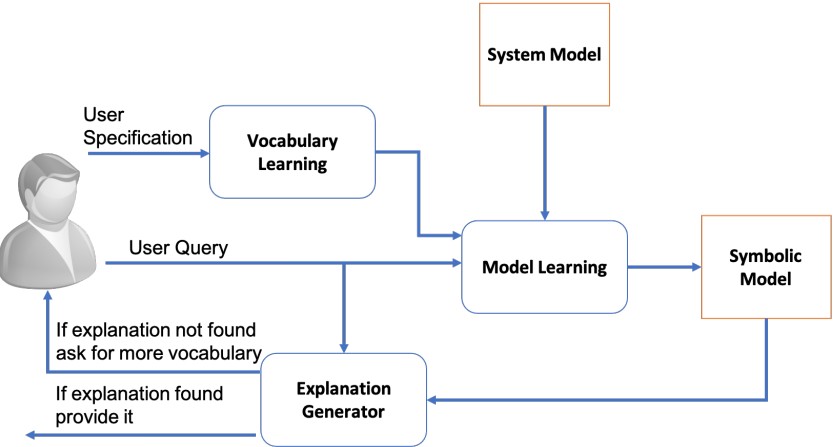

Figure 2: The diagrammatic representation of the various steps involved in the generation of post hoc symbolic explanations.

data collection strategies discussed below will assume that the user can visually observe the current state and agent actions. This doesn't necessarily mean the agent actually reasons about the world in terms of raw visual information. It may well be that the human can observe an embodied agent acting in the world or the agent can expose some visual representation of its internal state (like in the case of ATARI agents, where the agent presents a visual representation of its ram state).

**Actions**   The first obvious task would be to let the system identify the set of actions. This would be relatively straightforward in many RL tasks, including games where the action set of the original problem set is limited, and there exists a natural label of each of the possible actions. Though for many domains like robotics, the actual action space may be too complex (or even continuous) for the human to identify each action name. Instead, the human may think about the agent's action in terms of temporally extended actions (compared to the original actions). So instead of a series of joint angle changes, they may instead think in terms of abstract actions like picking and putting down objects. The person could communicate such abstract actions by labeling a set of agent demonstrations or by even providing demonstrations for each action. In either case, one could obtain a sequence of trajectories that correspond to each higher-level action, and one could learn classifiers that map trajectories to high-level action labels.

**Fluents**   The other vocabulary item of interest to us is the state fluents. This could consist of propositional or relational factors the user believes is relevant to the given task. In the case of propositional fluents, the user could specify the set of important fluents to the system by providing a set of states where the fluent is true and a set of states where the fluent is false. These examples could then later be used to train classifiers for each concept. For relational concepts, the user could start by labeling relevant objects and then, similar to the propositional case, provide positive and negative exam-

ples for each predicate of interest.

Note that a core flexibility provided by the setting is the fact that it allows the original vocabulary set provided by the user to be *incomplete*. This makes it a fundamentally different enterprise from all the other works that try to force the decision-making algorithms to use interpretable features (cf. (Koh et al. 2020) for single-step decision making, and (Lin, Lam, and Fern 2021) for sequential problems). On the one hand, these methods can guarantee that the system is considering these features, on the other, they are also inherently limited by the original vocabulary set. It can do no better than what is possible under the original vocabulary set. In contrast, under this method, the system is free to choose the best representation of the problem that allows it to come up with solutions efficiently. While this is still a widely debated topic, at least the dogma in mainstream RL seem to be that while symbolic representations are useful for end-users to understand and interact with the system, forcing the system to reason over human-engineered representations and knowledge would hamper the system in many practical scenarios as they preclude the use of more general and scalable methods (cf. (Sutton 2019; Silver et al. 2021)). In this case, vocabulary is only used for explanations. Additionally, given the fact that the system has access to the more complex internal model, the explanatory system is also able to tell when the given vocabulary set is insufficient to explain the given decision. Once the system has detected this, it can query the user for more vocabulary items. We would expect the system should to do it in a directed way, though performing directed concept acquisition is very much an open problem.

## 3.2   Model Learning

The next important aspect of the entire process is to take the vocabulary item and try to build a symbolic approximation of the overall model. There are multiple ways one could go about learning the models. One might be to generate a bunch of plan traces, use the learned vocabulary items to match into symbolic terms, and then use any of the existing model learning methods (cf. (Stern and Juba 2017)) to learn the final model. Alternatively, one could also employ a

more active learning process in which the agent actively interacts with the environment until it finds a model that meets the required criteria (this is similar to the strategy employed by Sreedharan et al. (2020)). Given all the existing work, we won't delve too much into the learning problem itself but rather look at some of the more unique possibilities that arise in this specific problem setting.

**Local Approximations:** While there aren't many explanation generation methods for sequential-decision making problems that look at post hoc explanations (particularly ones that try to build alternate models), post hoc representation learning is a very popular method in generating explanations for single shot decisions (Lakkaraju, Adebayo, and Singh 2020). A common technique used by many of these methods to simplify the explanations is to focus on creating local approximations of the original models. Popularized by Ribeiro, Singh, and Guestrin (2016), under this technique, rather than generating post hoc models that try to approximate the full model, they try only to capture the behavior of the model in a region of interest over the input space. Usually, this may correspond to data points close to the one that needs to be explained. We can translate the idea of local approximations also to our setting, where we can choose to learn a symbolic model that approximates the true model only for a subset of states and actions. The next natural question would be how to decide this set of actions. One approach would be to follow Ribeiro, Singh, and Guestrin (2016) and the earlier machine learning explanation works and choose distance as the deciding factor. In particular, consider only states within some distance from the initial state or the states in the current plan, and consider only actions that are possible in those states. A very natural distance measure for planning problems would be reachability or, in particular, reachability within a specified number of steps. Though rather than just blindly focusing on reachability/distance, one could also select the state and action subset more effectively if we are aware of the user's intentions for asking the query. For the previously discussed use case, if we restrict reachability to only states that are part of screen 1, the precondition becomes not_next_to_skull (as Joe can't acquire a sword in that screen).

**Learning Abstract Models vs Model Components:** Unlike the traditional use cases of learning planning models, here, we may not need to learn the entire model. Instead, depending on the explanatory query, we may need to only learn an abstract version of the model or even just identify parts of the model. For example, Sreedharan et al. (2020) looks at identifying explanations meant to refute alternate plans provided by the user. In this case, they identify only the required preconditions and an abstraction of the cost functions needed to refute the user queries.

**Post Hoc Explanatory Confidence:** Another important factor that is worth considering in this setting is measuring how accurately the learned model approximates the system model. Unless the explanatory system is exhaustively generating all possible transitions and behavior possible in the region of interest, there is a possibility that the learned model may not accurately reflect the actual behavior. If the model is wildly different from the true model, it could end up inducing incorrect beliefs in the end-user about the task and the system's understanding of the task. One way to try addressing such issues may be to provide the system with the ability to quantify its uncertainty about the model. Then it could use those measures to decide when it may be safe to provide explanations or even surface its uncertainty to the end-user. While there are some existing works on quantifying PAC guarantees for model learning (Stern and Juba 2017), this generally is an underexplored problem. Additionally, if the learned vocabulary mapping (from system's representation to user's vocabulary) is noisy, the symbolic traces that the explanation system collects may be incorrect and this should also be reflected in the confidence it assigns to the learned model. Sreedharan et al. (2020) presents some methods for creating such confidence measures under certain assumptions about the task.

**Reusing Previously Learned Model Components:** In the end, interacting with the complete model will be an expensive process and we would want to avoid performing this unless it is completely required. This means, being able to recognize cases where any new queries raised by the user can be resolved by a previously learned model representation. Also developing methods that are able to stitch together previously learned model components and abstractions to more complete model representations and checking if they suffice to address the user queries.

## 3.3 Explanation Generation

We won't delve too much into the exact explanation methods that could be used to generate the target explanations. But will bring up the two factors that may be worth considering in this question.

**Explanatory Queries:** As mentioned earlier the model learning is driven by the explanatory query. Keeping with most mainstream works in XAIP, we will assume most of these queries are contrastive in the sense that the user is trying to understand why a specific decision was made against a possible alternate decision the human was expecting. Though in this case, we have to make an additional level of distinction, namely what the explanation is trying to establish why the current decision is better than the alternative raised by the user, or why and how the system decided to make the decision. This was a distinction established in Langley (2019), where the author refers to the former as preference accounts and the latter as process accounts. The post hoc explanatory methods are particularly well suited for preference accounts. As they can be evaluated on these post hoc models independent of the original decision-making process used to derive the system's decisions.

**Identifying When the Model is Incomplete:** The next important feature required for the explanation generation, is to explicitly allow for the fact that the model being used to generate the explanation may be incomplete. So the explanation generation method needs to be able to identify cases where the current learned model may be incapable of generating the required explanation and as such the system needs to query the user to acquire new vocabulary items which may be used to augment the given model. Access to a system model could be extremely useful in such scenarios, particularly for evaluating user-specified alternatives in the context of contrastive queries. As the system model could be used to test the validity and cost of these alternatives.

## 4 Research Challenges and Opportunities

This section will discuss some of the big open questions and research opportunities posed by this direction. This is in addition to smaller problems like tailoring explanations to specific types of models that may be popular in fields like RL, considering stochastic models, etc.

**Acquiring new Concepts:** One of the open challenges is to address cases where the original vocabulary set is incomplete. The agent now needs to query the human to expand its vocabulary. One obvious strategy may be to ask for more concepts, though this could be a very inefficient way to collect more concepts as the ones the human may provide may be completely irrelevant to the given problem. An advantage the agent has is that it has access to its own representation of the task and thus may be able to provide some hints to the human as to what concepts may be relevant to the current query. One way to accomplish this may be to leverage low-level visual explanations (when a common visual channel is available). While we are unaware of any works in sequential-decision explanation that have leveraged such methods to collect concepts, a closely related work that has looked at collecting such concepts is (Hamidi-Haines et al. 2018), where they developed an interface that allows users to name certain regions of the state highlighted according to their relevance to the decision-making process. One would want to build on such interfaces for more general sequential decision-making systems and also possibly relax various assumptions like the fact that each concept corresponds to specific parts of the image. Another possibility is to revisit the end-to-end model learning methods similar to (Bonet and Geffner 2019), that also learns symbols. An open research question here is developing methods that check if any of the automatically discovered concepts or composition of such concepts could potentially map to concepts at the user's end. Additionally, we could also check if one could introduce inductive biases into these systems that allow the generation of naturally interpretable concepts ((Yeh et al. 2020) discussed how the assumption of the locality could be used in single-shot decisions).

**Incorporating Possible Noisy Decision-Making:** One of the questions that we have ignored in this paper is whether the agent's decision-making process can correctly use their internal model. Even in the best case, the symbolic models will only reflect what is present in the internal model. Though given the realities of the state of the art RL methods, in many cases, it is hard to guarantee that the decision-making processes can correctly use their own internal models to generate their decisions. If, in fact, these explanations are presented as the reasoning behind the agent's decisions (i.e., a process account per Langley (2019)), it could lead to the user forming incorrect beliefs about the agent's reasoning capabilities. It is still very much an open question how we could leverage the specifics of the agent's decision-making processes to select more precise explanations. Some initial strategies we could employ include performing additional tests like checking whether perturbation of some concept identified as part of the explanation in the current state leads to the system choosing a different action, or using the system's own internal representations (for example, looking at intermediate layer activations in the case of neural networks) to build concept classifiers, etc.

**Allowing For Collaboration:** We strongly believe that explanatory systems should be evaluated in the context of the overall application. A common use case for such systems may be scenarios where the user and the system are collaborating to come up with better solutions (as in the case of iterative planning (Smith 2012)). In the most general case, we would want this to be a bidirectional interaction wherein both the agent and the human can influence each other's beliefs on what constitutes an ideal solution. While there have been recent works on developing methods that allow people to specify preferences/objectives for agent behavior (c.f. (Illanes et al. 2020; De Giacomo et al. 2019)), they have generally focused only on developing interfaces to let the user specify constraints over the behavior the agent can generate using its internal models. For these systems to be truly successful, we need to not only allow the agent to provide explanations over why it may be performing certain behaviors, in terms of its beliefs about the model of the task, but also provide the user the ability to override the agent's belief about the task. We strongly believe that symbolic models can provide us with an interpretable interface to facilitate such bidirectional interactions. However, it's very much an open question on how to effectively take these post hoc model updates and fold them into the agent's decision-making process that may be using inscrutable models.

**Few Shot Learning of Concepts:** Unless the concepts are being pre-specified by a domain expert/designer where they could invest in collecting a large number of examples, these concepts would need to be learned from a few examples. If the agent is using learning methods that can compute its own representations of the state for coming up with decisions. Such simplified representations could then be used as the input to our vocabulary item classifiers.

**Teaching Concepts:** One of the possible scenarios we have not quite considered in this paper is what happens when

the human's vocabulary has no concept that could potentially explain the current decision. For example, there may exist no concepts that can describe the kind of patterns AlphaGo may be looking for to decide novel moves. In such cases, the agents would need to teach the human new concepts. We are unaware of any works that have even begun to look at these problems, though one can imagine effective solutions to this problem would need to make use of strategies from intelligent tutoring systems (ITS) (Anderson, Boyle, and Reiser 1985), natural language generation, and many other subareas of AI.

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
