# OpenReview forum: "Leveraging PDDL to Make Inscrutable Agents Interpretable: A Case for Post Hoc Symbolic Explanations for Sequential-Decision Making Problems"
_icaps-conference.org/ICAPS/2021/Workshop/XAIP — XAIP 2021_

### Official Review · AnonReviewer2 · 2021-07-01
**Timely and interesting position paper that will hopefully cultivate discussion of this topic at the XAIP workshop.**

**Rating:** 7
**Confidence:** 3

**Review:**

This paper argues that XAIP techniques can and should be leveraged to generate post-hoc explanations for sequential decision making systems that are typically inscrutable. Concretely, by learning symbolic models that approximate the decision making mechanism of the inscrutable system, one can apply the rich variety of XAIP techniques that have been proposed in recent years and generate meaningful explanations in response to user queries. The authors nicely enumerate the various components needed to realize this vision, and conclude by discussing research challenges and opportunities that arise from this proposed research direction.

This is a great paper to include in the workshop that will hopefully encourage discussion of this topic.

***Suggestions***

I appreciated the inclusion of the Montezuma example and I think it could be utilized even more extensively to ground the discussion further and benefit the reader. In terms of clarity, while it’s extremely challenging to cover a number of (seemingly disparate) research areas in a short paper, I found myself re-reading section 3 a number of times in order to not lose track of the overarching narrative. I believe a more extensive use of the example could help. Moreover, Figure 2 nicely captures the vision put forth by the authors and making additional references to it throughout the paper could help the reader establish relations between various ideas discussed in the paper.

***Typos***

**Intro**

Moreover, models like PDDL provides -> Moreover, models like PDDL provide

**Local approximation subsection**

if we restrict reachability to only state that are part - >, if we restrict reachability to only state**s** that are part

**Explanatory queries subsection**

why a specific decision was made against (a possible alternate decision the human was expecting). - > why a specific decision was made (against a possible alternate decision the human was expecting).  **Or rephrase some other way**

**Allowing for collaboration subsection**

For these systems to truly successful,-> For these systems to truly be successful (missing ‘be')

---

### Official Review · AnonReviewer1 · 2021-07-06
**Review: Post-hoc explanations**

**Rating:** 6
**Confidence:** 3

**Review:**

This challenge paper provides ideas on how explanations (within the realm of AI planning) can be generated if the decision making process is a black box (called inscrutable in the paper), i.e., if the plan generation cannot be investigated for providing explanations. In such cases, only the plan itself can be used, as no model is available (which could be learned, as proposed in the paper).

I had a hard time with the paper. If there were borderline, I would have chosen that. Sadly, there isn't, so I had the hard choice of slightly below or slightly above. I really wasn't sure about this and was on the "slightly below" side for quite a while, until I changed my mind because there's essentially nothing really "wrong" in it and it's only a workshop, I just don't see any strong (or clear) contributions. And maybe discussing the paper at the workshop will have some positive impact on the community, though I'd certainly don't fight for the paper in case the other reviewers are (even) more critical than I am.

My main critiques are that the paper is too abstract for my taste, though that might just be my personal preference, or even a consequence of the kind of paper (challenge). Another problem, related to the latter, is that I could not find a clear problem formulation, which seems to be very essential, even if you do not have clear solutions. Also the structure was unclear to me: What is a survey, what is a challenge, what is a proposed solution? I found that sometimes *really* confusing.


detailed comments: (some minor or even trivial (like language), some are more important)

- Although this is just a workshop I would have appreciated a bit more care regarding proof-reading the paper. There are more language errors than would be acceptable for a published paper, so that makes it sometimes unpleasant to read (see corrections below, but I'd appreciate not having to do that).

-abstract:
* "within ICAPS-community" should be "within the ICAPS-community"
* "Namely to help generate symbolic post hoc explanations [...]". I do not believe that this can be a sentence on its own. I think there is a verb missing. However, you can just correct it by not putting it into it's own (incomplete) sentence, but by appending it to the previous sentence, i.e., "..., namely to ...".
* "Through this paper, we hope to discuss how we could generate such post hoc explanations." -> This does not make any sense. You do not hope that you discuss this: Either you *do* discuss this in this paper or you don't, but since it already exisits you don't hope that there is a discussion in it. Maybe you wanted to say that you discuss how you hope how post hoc explanations could be generated? But even that sounds a bit odd, since it does not matter whether you hope that the proposed techniques work or not. No need to state such personal feelings, just state that you discuss techniques about potential explanation generations.
* "Motivate how one could use the current XAIP techniques [...] and also discuss [...]." This is incorrect English. It's missing something. You mean "We motivate". You can start a *part* of a sentence like this if the "we" is used in the beginning of the same sentence, but it's wrong (at least I strongly thing so) if it's only part of a previous sentence. (Which is the case here.)


- introduction/everywhere: Sometimes you use paper citations as objects within sentences, which is regarded bad style by many (me included, I find that very, very ugly). Note that most authorkits include commands to prevent exactly such citations. For this there are the commands shortcite and authorcite.
 * For example, instead of writing "While there are some works like (Sreedharan et al. 2020) that have..." you should write "While there are some works like the one by Sreedharan et al. (2020) that have...".
 * E.g., instead of "Following (Sreedharan et al. 2020), we will...", you should write "Following Sreedharan et al. (2020), we will...". The original refers to the paper (which is ugly because in 95% of the cases such citations are not objects within sentences, they are just added without being an object ot be actively read), whereas the latter refers tp the people (i.e., Sreedharan et al.).
 * E.g., "For example, (Sreedharan et al. 2020) looks at" would become "For example, Sreedharan et al. (2020) look at" (Note that looks changed into look, since it's now referring to multiple people instead of to a single paper.)
 * there are many examples like this in the paper. If you decide to change the style, please do it everywhere to stay consistent.

- intro: "models like PDDL provides a very intuitive representation" -> "provide" (plural)

- in the introduction you state that "[in] this paper, we hope to make a systematic case for applying and adapting existing methods for both explanation **and model learning** to this new setting". However, the abstract seems to mention only one of these two contributions, which is odd. If you have these two main contributions, both should be mentioned in the abstract.

- You do not introduce/explain RL (Sec. 2)

- "a plan that involves the Panama-Joe". The definite article sounds very strange (i.e., wrong) here. I'd remove the "the".

- Since this paper is a challenge paper in the field of explainable planning, doesn't it make sense to cite the recent survey on that topic ("The Emerging Landscape of Explainable Automated Planning & Decision Making", IJCAI 2019). Given that both authors are co-authors of that paper I was rather surprised that it wasn't cited. The very first sentence sounds like an excellent position for that citation.

- "Let us imagine, an observer who is not familiar with all the rules of the game is" -> the comma should be removed.

- Sec.2, "we will use Montezuma’s revenge [...] as an example to illustrate the potential utility of such explanations." -> This sentence appears in the first sentence of Section 2. Thus, no explanations have been mentioned by then. Therefore, it's odd to mention "such explanations", when it's not clear (within that section" what that such is referring to. You probably mean "post-hoc symbolic explanations" since that's what the paper is about, but language-wise that is still odd, so you should qualify which explanations you mean.

- "a plan that involves the Panama-Joe (the protagonist of the game)" the article (the Panama-Joe) is wrong here, because Panama-Joe is not known yet at this point, so its just a name, which therefore does not have an article.

- The first rather large paragraph of Section 3 is a good example for me being lost. No matter how often I read it, I still don't know what's happening here. The first odd thing is that this appears to be a challenge paper, so I'd assume that you explain the challenge first. Is that Sec. 3, or was the challenge already completely done by Section 2, the "case study"? It looks like Sec. 3 is your proposed solution, but you actually do not say so.
If Sec. 2 is supposed to completely introduce the challenge, you'd benefit from calling the section that way. If Section three shows your hypothesized solutions, maybe call it that way too. Anyway, you start by explaining an architecture. What is that? It's probably not from the literature because there is no citation. Is it your proposed solution to "the challenge"? If so, you should say so. I also did not understand what you mean by "vocabulary items" and "explanation query". There is actually no explanation for this, yet you use it in that paragraph as if they were clear (which they are not). E.g., just two sentences later you write "With the vocabulary set and the query in place, the explanation generation procedure can interact with the actual system to generate a symbolic approximation that is sufficient to provide a response to the current query." But I have no idea what that means with none of the concepts being introduced. Even if I knew somehow what you mean by vocabulary and query, what is that symbolic approximation that you talk about here for the first time? This is really absolutely unclear.
Let me stress that parts of that become clear after one has read 3.1, but while still being before this, it is not clear yet. (And it remains unclear, for example, what the query is.)

- You use "c.f (paper)" twice. Apart from the fact that this is another instance of using paper citations as objects within sentences (which I argued against), the abbreviation is wrong. It should be "cf." instead. You also use "c.f." once, which does not make sense because that's latin for "conferatur", which is a single word, not two.

- "While there aren’t many explanation generation methods for sequential-decision making problems that look at post hoc explanations (particularly ones that try to build alternate models)." -> This sentence is incorrect, something is missing here. E.g., "While I am bored." is clearly incorrect, whereas "While I am bored, my heart beats slowly." would be correct. Note that "While I am bored. My heart beats slowly." is *also* wrong for the same reason: The second sentence is a complete sentence, but the first is not. I say this, because I suspect that you did not forget anything, but just wanted to append the next sentence to it. The same flawed construction is given later, starting with "While there have been recent works..."

- The first paragraph in Sec. 3.3 is missing its dot at the end.

- "is trying to establish, why the current decision" -> the comma should be removed

- "we will assume most of these queries are contrastive in the sense that the user is trying to understand why a specific decision was made against(a possible alternate decision the human was expecting)." -> The parenthesis appears wrong. It means that it's content is bonus information, doesn't it? But it's not! If you remove the entire parenthesis including its content, the sentence is not correct anymore.

- "in (Langley 2019), where they refer to" -> the "they" is wrong since here is only one author. Also, you are breaking with your own systematicity here, because you treat (Langley 2019) as a *paper* not as their authors. So according to your citation style (that I argue against) you would have to write "in (Langley 2019), where it refers to". But as said, I would not treat citations as objects in sentences anyway. So according to my style you would write "by Langley (2019), where he refer to" (If Langley is a woman you had to write "she" of course. Or you write "the author" instead.)

- "For these systems to truly successful" -> "to be truly" ("be" is missing)

---

### Meta-Review · Area_Chairs · 2021-07-07

**Recommendation:** Accept
**Confidence:** 5

**Metareview:**

We thank the authors for their contribution. The reviewers agree that the paper would facilitate interesting discussions during the workshop. Please refer to the comments/concerns provided by the reviewers when preparing your camera-ready submission.

In particular, the paper would benefit from a more clear demarcation of the problem being addressed, with a special focus on Section 3, which happened to be rather hard to follow. For example, what is the formal problem formulation you wish to solve? A more grounded representation of the ideas considered would improve the clarity of the paper and strengthen your arguments. As reviewer 2 suggests, you could also consider referencing Figure 2 throughout the paper in order to help the readers relate your proposed ideas with respect to the various stages involved in generating post hoc (symbolic) explanations. Finally, reviewer 1 pointed out fixes to language/typo errors made in the paper; As they affect the legibility of the paper, please correct them for the camera-ready submission.

We are looking forward to an interesting and fruitful discussion at the workshop.

---

### Decision · Program_Chairs · 2021-07-08

Accept